# Not-So-Sweet Dreams: Plasma and IgG N-Glycome in the Severe Form of the Obstructive Sleep Apnea

**DOI:** 10.3390/biom13060880

**Published:** 2023-05-23

**Authors:** Doris Plećaš, Nikol Mraz, Anne Marie Patanaude, Tea Pribić, Ivana Pavlinac Dodig, Renata Pecotić, Gordan Lauc, Ozren Polašek, Zoran Đogaš

**Affiliations:** 1Mediterranean Institute for Life Sciences, 21000 Split, Croatia; doris.plecas@gmail.com; 2Genos Glycoscience Ltd., 10000 Zagreb, Croatia; nikolmraz@genos.hr (N.M.); anne-mariepatenaude@genos.hr (A.M.P.); tpetrovic@genos.hr (T.P.); glauc@genos.hr (G.L.); 3Department for Neuroscience, School of Medicine, Sleep Medicine Center, University of Split, 21000 Split, Croatia; ivana.pavlinac@mefst.hr (I.P.D.); renata.pecotic@mefst.hr (R.P.); 4Faculty of Pharmacy and Biochemistry, University of Zagreb, 10000 Zagreb, Croatia; 5Department of Public Health, School of Medicine, University of Split, 21000 Split, Croatia; ozren.polasek@mefst.hr; 6Department of General Courses, Algebra University, 10000 Zagreb, Croatia

**Keywords:** N-glycome, sleep, biomarker, aging

## Abstract

Obstructive sleep apnea (OSA) is a prevalent disease associated with increased risk for cardiovascular and metabolic diseases and shortened lifespan. The aim of this study was to explore the possibility of using N-glycome as a biomarker for the severe form of OSA. Seventy subjects who underwent a whole-night polysomnography/polygraphy and had apnea–hypopnea index (AHI) over 30 were compared to 23 controls (AHI under 5). Plasma samples were used to extract 39 glycan peaks using ultra-high-performance liquid chromatography (UPLC) and 27 IgG peaks using capillary gel electrophoresis (CGE). We also measured glycan age, a molecular proxy for biological aging. Three plasma and one IgG peaks were significant in a multivariate model controlling for the effects of age, sex, and body mass index. These included decreased GP24 (disialylated triantennary glycans as major structure) and GP28 (trigalactosylated, triantennary, disialylated, and trisialylated glycans), and increased GP32 (trisialylated triantennary glycan). Only one IgG glycan peak was significantly increased (P26), which contains biantennary digalactosylated glycans with core fucose. Patients with severe OSA exhibited accelerated biological aging, with a median of 6.9 years more than their chronological age (*p* < 0.001). Plasma N-glycome can be used as a biomarker for severe OSA.

## 1. Introduction

Glycosylation is one of the most widespread and functionally diverse co- and posttranslational protein modifications [1]. Unlike proteins, glycans are not directly encoded in the genome; instead, their regulation is controlled by several hundreds of enzymes, transcription factors, and other proteins [1,2,3,4], which account for as much as 1–4% of the human genome [5]. Although most glycans cover the surface of cells [6], there is also a free-floating fraction of glycans in the plasma [7], which can be used as biomarkers for disease risk and monitoring assessment [8,9].

In addition to genes, the composition of glycans is influenced by environmental factors, making glycans good indicators of the combined effects of environment and genes on human physiology [10]. Consequently, glycans were implied in numerous physiological and pathological conditions, including cancer [11,12], autoimmune [13], and infectious diseases [14]. A comprehensive study reported that glycans are directly involved in the pathophysiology of major diseases and that additional knowledge from glycoscience will be required to achieve the goals of personalized medicine [15]. Interestingly, glycans have also been implied as a good biomarker for aging, reflecting various processes with strong age-related patterns [8].

Obstructive sleep apnea (OSA) is a common disorder in the general population, particularly in men, characterized by recurrent episodes of partial or complete upper airway obstruction during sleep followed by intermittent hypoxia and increased sympathetic nervous system activity [16]. OSA has been associated with an increased risk of cardiovascular, metabolic, and inflammatory diseases, neurocognitive decline, and overall mortality [17,18,19,20,21,22,23]. Furthermore, OSA was independently associated with lipid abnormalities, insulin resistance [24], and visceral obesity [25]. The diagnosis of OSA is performed in specialized sleep labs or sleep medicine centers through whole-night polysomnography (PSG) or polygraphy (PG). However, in terms of the accessibility of sleep medicine services and the high prevalence of OSA among the general population, diagnosis can sometimes be challenging [26]. Thus, there is a constant need to find a reliable biomarker for diagnosing OSA. Current OSA screening tools rely on questionnaires with low diagnostic accuracy [27] and blood biomarkers [28,29,30,31,32]. These biomarkers include glycated hemoglobin (HbA1c), C-reactive protein (CRP), erythropoietin (EPO), interleukin-6, uric acid, endocan, copeptin, and numerous other molecules involved in the pathogenesis of neurodegenerative diseases, lipid, and vascular metabolic pathways [29,30,31,33,34,35,36,37,38]. Chronic inflammation, hypoxemia, sleep fragmentation, and stress induced by OSA are associated with biomarker changes. Still, the diagnostic utility of individual biomarkers or combinations of markers is inconclusive in a priori identifying OSA [29]. Considering the importance of the inflammatory component of OSA, several studies have been conducted focusing on multiple inflammatory biomarkers simultaneously. Flaming et al. showed that the most predictive individual OSA biomarkers were CRP + HbA1c + EPO [39], replicating a previous feasibility study [40]. The inflammatory response, which plays a role in the pathophysiology of OSA, requires fine-tuning many cellular and extracellular processes, such as complement activation. The complex arrangement and highly precise regulation of these molecular mechanisms depend, among other things, on the correct N-glycosylation of the molecules [2].

The aim of this study was to explore the possible role of plasma and IgG N-glycome patterns as the biomarkers of OSA by comparing a group of severe OSA patients with healthy controls. Furthermore, we calculated glycan age, a molecular proxy of biological ageing, and compared it among patients with OSA and controls.

## 2. Materials and Methods

### 2.1. Study Setting and Patients

This study was performed at the Sleep Medicine Center Split, University of Split School of Medicine and Clinical Hospital Split, Croatia. The Ethical Board of the University of Split School of Medicine in Split approved the study. The study included 93 consecutive subjects attending the Split Sleep Medicine Center of the University Hospital Split in 2022. The inclusion criteria were adults (over 18 years of age), without severe disability or a known medical condition requiring institutional care. All subjects were informed of the study protocol, objectives, and potential risks and enrolled only after signing a written informed consent form.

### 2.2. Measurements

The diagnostic procedure was the same for all the patients, starting with a medical history and a short physical examination. Next, all patients underwent polysomnography/polygraphy (PSG/PG) testing. The results of either the whole-night PG (SOMNOcheck2, Weinmann, Hamburg, Germany) or PSG (Alice 6, Philips Respironics, Eindhoven, the Netherlands; Alice NightOne, Philips Respironics, Eindhoven, the Netherlands) were scored and interpreted according to AASM diagnostic criteria and ESRS guidelines [41]. A certified sleep technician and physician evaluated OSA. The diagnosis and severity classification of OSA was based on the apnea–hypopnea index (AHI), with AHI ≥ 30 indicating severe form and AHI < 5 indicating no OSA. Following the whole-night PSG/PG, venous blood samples were taken from all patients.

For purposes of this study, we selected only two subgroups. The first one included patients who had AHI over 30 and were considered to have severe disease. The second group consisted of subjects with negative polysomnographic testing; this group was considered a control. In addition to polysomnography, all subjects provided a blood sample. Blood was collected into 4 mL EDTA vacutainers. Samples were centrifuged within an hour of extraction (3000 rpm, 15 min, +4 °C). Plasma was separated from the cells, divided into 100 µL aliquots and stored at −80 °C.

### 2.3. Plasma Glycan Measurements

Plasma glycans were measured by hydrophilic interaction ultra-high-performance liquid chromatography with fluorescence detection (HILIC-UHPLC-FLR). In short, individual plasma samples (10 μL) were denatured by incubation with SDS (Invitrogen, Waltham, MA, USA) at 65 °C for 10 min, followed by the release of glycans with the addition of 1.2 units of peptide: N-glycosidase F (Promega, Madison, MI, USA) and overnight incubation at 37 °C [42,43,44]. Next, N-glycans were labelled with 2-aminobenzamide (Sigma-Aldrich, Saint Louis, MO, USA), separated by hydrophilic interaction liquid chromatography on an Acquity Ultra-Performance Liquid Chromatography (UPLC Technology) instrument (Waters, Milford, MA, US), and quantified with a fluorescence detector. Data were processed using an automated integration method [45], and chromatograms were separated into 39 glycan peaks (GPs), corresponding to a single molecule or a mixture of several structurally similar molecules (the list of all glycan structures underlying the detected peaks is provided in the Appendix A).

Block randomization was used to determine the position of samples in 96-well plates containing approximately 70 samples, with five randomly selected technical replicate samples from the same plate and five from other plates. In addition to the sample replicates, four internal plasma standards were included in each plate to maintain quality control and allow batch correction to be performed later.

### 2.4. IgG Glycan Measurements

IgG glycans were measured by capillary gel electrophoresis with laser-induced fluorescence (xCGE-LIF) [46]. This process resulted in 27 IgG glycan peaks (the entire list and the corresponding structures for these peaks are provided in the Appendix A). The analytic protocol was based on the previous study [47], relying on the release and fluorescent labelling of N-glycans from previously isolated IgG. Isolation of IgG from blood plasma was performed by utilizing a Protein G monolithic plate, in line with the previously described protocol [46]. All buffers must be prepared or stored at 4 °C (except 10× PBS, pH 7.40) for up to 7 days. Buffers were filtered through 0.2 µm PES filters (Nalgene brand, Thermo Fisher Scientific, Waltham, MA, USA). The resistance of used ultrapure water was maintained above 18.2 M at 25 °C.

N-glycans were released from denatured glycoproteins in the presence of PNGase F enzyme. Further, released N-glycans were labelled with 8-aminopyrene-1,3,6-trisulfonic acid trisodium salt (APTS) and purified from salt and excess label by HILIC-based solid phase extraction (HILIC-SPE). The glycoprofiling was performed using a DNA sequencer (3130 Genetic Analyzer, Applied Biosystems, Waltham, MA, USA) and analyzed by the multiplexed capillary gel electrophoresis with laser-induced fluorescence (xCGE-LIF) method. Data processing and analysis were performed by glyXtool software (glyXera) [46].

### 2.5. Statistics

Means and standard deviations were calculated for all numerical variables (with the median for the glycan age estimation due to large variance), whereas categorical variables were reported as numbers and percentages. The categorical data were tested with the chi-square test, and the numerical data were tested with the *t*-test. Due to non-normal data distribution (based on the results of the Kolmogorov–Smirnov test), the comparison of chronological and biological age was based on Wilcoxon signed rank test. The final step of the analysis was based on linear regression that predicted AHI, in which significant predictors from the bivariate analysis were included, with age and sex as mandatory predictors. Furthermore, to elucidate whether the BMI moderated the association of glycans with AHI, we made two additional models stratified by the body mass index (BMI). A BMI value of 25 kg/m^2^ was used as the cut-off point, with BMI being excluded as a confounder in these models. In addition, we calculated glycan age, a proxy for biological ageing. The calculation was based on the approach from the original paper [8] and further adjusted to better fit the data in this study. Firstly, we used linear regression to model the relationship between IgG glycans and chronological age only in controls and select the variables used in the age estimation. Three glycans were used in the final model, namely peaks 15, 18, and 26; notably, three selected glycans correspond to UPLC peaks used in the previous analysis (P15 corresponds to the GP4 peak, P18 to GP6, and P26 to GP14, all used in the original glycan age formula). The model had a kappa coefficient between chronological and biological age of 0.8, indicating perfect agreement. The same formula was applied to cases to obtain their glycan age estimates. All analyses were performed in R, with significance set at *p* < 0.05.

## 3. Results

The study included 93 subjects who underwent polygraphic/polysomnographic OSA diagnostic procedure and were split into two groups based on their AHI value—cases (*n* = 70; AHI over 30) and controls (*n* = 23; AHI up to 5). The initial comparison revealed significant differences in age, BMI, AHI, and the prevalence of hypertension in medical history (Table 1).

Analyzing plasma glycans, we observed increased values of GP1, GP2, and GP32 and decreased values of GP8, GP10, GP18, GP24, and GP28 in patients with the severe form of OSA (Table 2). In the case of IgG glycans, significant increases were observed in P26 and P27, while P9, P14, P15, and P18 were decreased (Table 3).

The next analytic step was based on a linear regression model, with significant glycan peaks from the bivariate analysis. The regression model for plasma peaks suggested that GP24, GP28, and GP32 were independent predictors of AHI in a model that controlled for multiple confounding effects (Table 4). Stratification according to BMI indicated no significant results in subjects with body mass index under 25 kg/m^2^, while two out of three glycans from the full model remained significant (Table 4). A similar analysis plan for IgG glycans indicated that only one peak was significant for the entire model, P26; these effects were not seen in stratified analysis (Table 5).

The glycan age estimation was firstly based on the controls used to validate and develop the sample-specific formula. The final formula was based on three glycan peaks, with the following coefficients: glycan age = 25.159 + (1.093 × P15) + (5.535 × P18) + (−1.125 × P26). Next, this calculation was used on OSA patients, yielding a strong difference in chronological and biological age, with an average of 9.2 years more than the chronological age in cases. However, due to the substantial deviation from normality, we calculated the median value, which suggested that severe OSA patients were 6.9 years older than their chronological age (*p* < 0.001; Table 6).

## 4. Discussion

The results of this study show that three plasma glycan peaks were significant predictors of the AHI index in the severe form of obstructive sleep apnea compared with controls. Among the four significant glycans peaks, two were decreased and two were increased in patients with severe OSA. The decreased ones were GP24, corresponding to disialylated triantennary glycans as a major structure (A3G3S2) and GP28, trisialylated triantennary glycans as a major structure (A3G3S3). The most interesting result, which retained the strongest significance level in the multivariate analysis, was an increased GP32 peak, corresponding to a trisialylated triantennary glycan as a major structure (A3G3S3). Interestingly, the molecular structures underlying these peaks are quite similar, since all three peaks contain triantennary glycans, suggesting that these do not differentiate the inverse relationship. The only structural difference between them was the addition of sialic acid in the minor structure; the first two peaks contained monosialylated and disialylated, with some trisialylated glycans, while the GP32 contained exclusively trisialylated glycans.

Previous studies have often linked GP32 with several chronic diseases, including diabetes mellitus, associated renal pathogenesis phenotypes, and insulin resistance [48,49,50,51]. Increased GP32 levels were observed in the pathogenesis of inflammatory and malignant liver disease and inflammatory bowel disease [52,53]. Additionally, the GP32 was associated with an increased triacylglyceride synthesis in breast carcinoma [54], hyperlipidemia [55], and several traits related to metabolic syndrome, including systolic and diastolic blood pressure, fasting glucose, and body mass index [56]. These findings suggest that GP32 is associated with chronic inflammation, one of the main pathogenetic mechanisms involved in OSA [30]. The important element in this analysis is the relationship of GP32 with obesity. For this purpose, we stratified the sample by BMI to better understand the patterns. The results have shown that the relationship between GP32 and AHI was stronger in overweight and obese patients compared to the normal weight patients. A possible explanation of these results may be a non-linear association of glycans with BMI. Therefore, a suggestion for future studies would be to include multiple data collection points to better understand the dynamics of changes across the severity of the disease.

The IgG fraction P26 was the only significant result, suggesting that the extent of differences was less strongly present in IgG than plasma. This is possibly due to a more dynamic nature of IgG, while plasma patterns have to result from more numerous and systematic effects, thus being less prone to quick changes and reflecting a stable pattern across various conditions [55].

Our results suggest that patients with severe OSA are experiencing accelerated ageing, as estimated by glycans. It has been shown that patients with severe OSA are indeed experiencing accelerated biological aging and that they have shortened lifespan [57]. However, several other studies suggested that comorbidities in patients with severe OSA may have a more important role [58,59]. Furthermore, it has been shown that mild apnea may even be linked to improved survival [60], suggesting the possibility of adaptive hormetic effects of sleep apnea.

The limitations of this study include a relatively small sample size, especially in controls, which is a direct consequence of methodological difficulties. Commonly, only the more severe presentations are sent to polysomnography; therefore, including healthy controls presents a challenge. Furthermore, the direct consequence is a possibly underpowered study to properly explore some of these patterns, including the relationship between OSA and obesity. The study further suffers from the survivor bias, which may be causing an underestimation of the true effect (due to early losses of patients with the most severe forms of OSA, who do not contribute to the oldest age group), as described in previous studies [18].

The results of this study suggest that IgG and plasma N-glycome may be considered biomarkers for severe OSA. Notably, the observed differences were stronger in plasma. Nevertheless, independently replicating these findings and incorporating traditional sleep-related biomarkers would provide additional insight into this complex interaction between metabolism and sleep. Alternatively, performing an interventional study on IgG and plasma N-glycome, where CPAP would treat OSA, would provide additional information on whether glycans might be used as suitable biomarkers for disease progress monitoring.

## Figures and Tables

**Table 1 biomolecules-13-00880-t001:** The initial comparison of cases and controls.

Variable	Cases (*n* = 70)	Controls (*n* = 23)	*p*
Sex; *n* (%)			
Men	59 (84.3)	20 (87.0)	0.756
Women	11 (15.7)	3 (13.0)	
Age (years); mean ± SD	57.2 ± 12.4	48.7 ± 13.7	0.006
AHI; mean ± SD	46.39 ± 20.10	2.71 ± 1.20	<0.001
BMI; mean ± SD	32.40 ± 6.00	25.70 ± 3.18	<0.001
Medical history			
Hypertension; *n* (%)	38 (57.6)	5 (22.7)	0.005
Type 2 diabetes mellitus; *n* (%)	10 (15.4)	2 (9.1)	0.459
Depression; *n* (%)	2 (3.1)	0 (0)	0.405
Asthma; *n* (%)	4 (6.2)	0 (0)	0.234
GERD; *n* (%)	14 (23.7)	2 (9.5)	0.162

**Table 2 biomolecules-13-00880-t002:** Comparison of plasma glycan peaks between cases and controls.

Plasma Glycan Peaks; Mean ± SD	Cases (*n* = 70)	Controls (*n* = 23)	*p*
GP1	5.67 ± 2.06	4.64 ± 1.14	0.003
GP2	2.20 ± 0.56	2.02 ± 0.29	0.049
GP3	0.09 ± 0.03	0.09 ± 0.02	0.412
GP4	3.75 ± 0.80	3.92 ± 0.74	0.387
GP5	2.21 ± 0.57	2.30 ± 0.51	0.522
GP6	1.34 ± 0.48	1.31 ± 0.27	0.781
GP7	1.17 ± 0.51	1.15 ± 0.23	0.896
GP8	1.44 ± 0.20	1.54 ± 0.21	0.036
GP9	0.17 ± 0.53	0.12 ± 0.02	0.613
GP10	3.27 ± 0.89	3.90 ± 0.69	0.002
GP11	0.85 ± 0.53	0.89 ± 0.21	0.699
GP12	0.91 ± 0.18	0.97 ± 0.17	0.155
GP13	1.13 ± 1.55	0.90 ± 0.18	0.494
GP14	13.01 ± 1.9	13.47 ± 1.12	0.276
GP15	0.49 ± 0.60	0.42 ± 0.07	0.566
GP16	4.90 ± 1.16	5.15 ± 0.59	0.320
GP17	1.91 ± 1.25	1.97 ± 0.74	0.838
GP18	2.82 ± 0.49	3.37 ± 0.38	<0.001
GP19	1.10 ± 0.14	1.08 ± 0.10	0.528
GP20	26.46 ± 2.95	25.94 ± 1.50	0.268
GP21	0.46 ± 0.42	0.42 ± 0.04	0.598
GP22	3.89 ± 0.85	3.75 ± 0.64	0.449
GP23	1.98 ± 0.71	1.80 ± 0.44	0.263
GP24	1.55 ± 0.42	1.80 ± 0.53	0.025
GP25	0.29 ± 0.29	0.27 ± 0.03	0.736
GP26	1.83 ± 0.38	1.74 ± 0.38	0.318
GP27	1.07 ± 0.31	1.12 ± 0.41	0.640
GP28	0.58 ± 0.17	0.66 ± 0.18	0.047
GP29	0.27 ± 0.73	0.17 ± 0.04	0.525
GP30	4.15 ± 1.29	4.44 ± 1.17	0.342
GP31	0.39 ± 0.23	0.36 ± 0.11	0.594
GP32	1.70 ± 0.44	1.33 ± 0.22	<0.001
GP33	3.25 ± 0.95	3.29 ± 1.09	0.881
GP34	0.36 ± 0.08	0.33 ± 0.06	0.069
GP35	0.42 ± 0.12	0.42 ± 0.15	0.826
GP36	0.57 ± 0.10	0.58 ± 0.08	0.636
GP37	0.42 ± 0.14	0.46 ± 0.09	0.208
GP38	0.96 ± 0.19	0.98 ± 0.13	0.609
GP39	0.95 ± 0.31	0.96 ± 0.33	0.984

**Table 3 biomolecules-13-00880-t003:** Comparison of IgG glycan peaks between cases and controls.

IgG Glycan Peaks; Mean ± SD	Cases (*n* = 70)	Controls (*n* = 23)	*p*
P1	0.30 ± 0.08	0.33 ± 0.20	0.461
P2	0.25 ± 0.05	0.26 ± 0.09	0.569
P3	1.49 ± 0.37	1.40 ± 0.39	0.323
P4	1.88 ± 0.34	1.78 ± 0.36	0.241
P5	0.16 ± 0.02	0.15 ± 0.03	0.182
P6	0.05 ± 0.02	0.06 ± 0.03	0.170
P7	0.24 ± 0.03	0.26 ± 0.05	0.167
P8	2.22 ± 0.33	2.33 ± 0.42	0.284
P9	0.32 ± 0.04	0.38 ± 0.09	<0.001
P10	0.65 ± 0.15	0.69 ± 0.15	0.272
P11	0.29 ± 0.05	0.27 ± 0.06	0.367
P12	8.54 ± 1.65	7.92 ± 1.79	0.149
P13	2.60 ± 0.55	2.37 ± 0.47	0.054
P14	0.55 ± 0.11	0.63 ± 0.16	0.034
P15	18.65 ± 2.8	25.38 ± 5.84	<0.001
P16	0.62 ± 0.26	0.64 ± 0.34	0.831
P17	0.43 ± 0.14	0.43 ± 0.16	0.952
P18	3.99 ± 1.07	4.83 ± 1.60	0.021
P19	0.39 ± 0.09	0.43 ± 0.12	0.089
P20	0.45 ± 0.08	0.45 ± 0.08	0.742
P21	18.46 ± 1.47	17.66 ± 1.73	0.051
P22	12.0 ± 1.24	11.36 ± 1.49	0.066
P23	6.01 ± 1.11	5.82 ± 1.02	0.452
P24	0.76 ± 0.13	0.78 ± 0.11	0.453
P25	0.18 ± 0.04	0.17 ± 0.04	0.098
P26	16.84 ± 2.51	11.71 ± 3.46	<0.001
P27	1.69 ± 0.27	1.49 ± 0.34	0.013

**Table 4 biomolecules-13-00880-t004:** A linear regression model with age, sex, BMI, significant predictors of AHI from bivariate analysis, and plasma glycans in three models; the full model (all cases with valid data included), in subjects with body mass index under or equal to 25, and BMI over 25 kg/m^2^.

	Full Model (*n* = 88)			BMI under 25 kg/m^2^ (*n* = 13)			BMI over 25 kg/m^2^ (*n* = 75)		
	Beta	t	*p*	Beta	t	*p*	Beta	t	*p*
Gender	0.07	0.67	0.503	3.05	0.69	0.614	0.04	0.37	0.711
Age	0.15	1.62	0.109	0.42	0.31	0.809	0.14	1.20	0.236
BMI	0.44	3.68	<0.001	*	*	*	*	*	*
Hypertension	−0.19	−2.02	0.047	1.11	0.47	0.719	−0.15	−1.30	0.200
GP1	−0.05	−0.41	0.680	1.52	0.79	0.574	0.29	2.00	0.050
GP2	0.15	1.17	0.246	−2.28	−0.71	0.607	−0.08	−0.53	0.595
GP8	0.00	0.04	0.965	3.97	0.57	0.669	0.15	1.23	0.225
GP10	0.06	0.56	0.580	0.69	1.38	0.399	0.08	0.59	0.556
GP18	−0.10	−0.74	0.461	−5.09	−0.72	0.601	0.01	0.04	0.967
GP24	−0.60	−2.21	0.030	−5.01	−0.59	0.662	−0.83	−2.34	0.022
GP28	0.48	2.09	0.040	4.95	0.64	0.638	0.55	1.92	0.060
GP32	0.50	3.28	0.002	1.64	0.55	0.682	0.79	3.85	<0.001

* BMI was omitted from stratified models.

**Table 5 biomolecules-13-00880-t005:** A linear regression model with age, sex, BMI, significant predictors of AHI from bivariate analysis, and IgG glycans in three models; the full model (all cases with valid data included), in subjects with body mass index under or equal to 25, and BMI over 25 kg/m^2^.

	Full Model (*n* = 88)			BMI under 25 kg/m^2^ (*n* = 13)			BMI over 25 kg/m^2^ (*n* = 75)		
	Beta	t	*p*	Beta	t	*p*	Beta	t	*p*
Gender	−0.10	−1.07	0.286	0.16	0.51	0.644	−0.17	−1.40	0.166
Age	0.03	0.25	0.802	0.44	1.24	0.303	0.03	0.18	0.858
BMI	0.53	4.94	<0.001	*	*	*	*	*	*
Hypertension	−0.03	−0.28	0.782	−0.25	−1.11	0.350	0.09	0.77	0.445
P9	0.12	1.01	0.316	0.96	1.97	0.143	0.19	1.19	0.239
P14	−0.08	−0.77	0.443	−0.90	−3.11	0.053	−0.08	−0.66	0.512
P15	−0.18	−0.64	0.527	−1.84	−0.97	0.403	0.42	1.33	0.187
P18	−0.02	−0.16	0.877	−0.36	−0.81	0.477	−0.21	−1.06	0.295
P26	−0.48	−2.37	0.020	−1.93	−1.69	0.189	−0.16	−0.67	0.506
P27	−0.02	−0.18	0.862	−1.28	−1.32	0.280	0.05	0.28	0.783

* BMI was omitted from stratified models.

**Table 6 biomolecules-13-00880-t006:** The comparison of age measures in cases and controls.

Variable	Cases (*n* = 70)	Controls (*n* = 23)
Chronological age (years); mean ± SD	57.2 ± 12.4	48.7 ± 13.7
Glycan age (years estimation); mean ± SD	66.5 ± 16.5	48.7 ± 9.6 *
Difference (years); mean ± SD	9.2 ± 17.7 (median 6.9, interquartile range 27.6)	*
*p* (chronological vs. biological age; Wilcoxon signed rank test)	*p* < 0.001	*

* Controls were used in the model generation.

## Data Availability

Data is available upon reasonable request to the corresponding author.

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
