# Peer review of "Not-So-Sweet Dreams: Plasma and IgG N-Glycome in the Severe Form of the Obstructive Sleep Apnea"

_biomolecules, 2023, doi:10.3390/biom13060880_

Round 1

Reviewer 1 Report

The authors present a study of plasma and IgG N-glycosylation in patients with obstructive sleep apnea (OSA).

The authors have a well-known expertise on glycomic studies and their interpretation and the topic is worthy of interest, so the manuscript deserves publication but only after some major modifications.

Major points.

The authors report differences in serum and IgG glycosylation and a significative increase in glycan age between patients affected by OSA and the control group. Since patients with OSA also have a high BMI it is important to know if there is a contribution of BMI in the differences and in the glycan age calculation. Authors also point out in the discussion that obesity may play a role and that some biomarker such as GP32 are significantly increased both here and in obese patients.

Since the authors have already published N-glycosylation studies on patients with high BMI in the past, a comparison between this group of patients and a group with only a high BMI should be included in this study. It should not be a problem for the authors.

Minor changes

In order to better understand the study performed, a table with the glycan structures and the various chromatographic peaks should be presented. It is true that the authors have presented this table in many of their works, but this paper must be able to be understood without having to search the entire bibliography. At least one table illustrating structures that have significant differences should be added.

Author Response

Dear Reviewer, 

please see the attachment that provides a point-by-point response to your comments.

Thank you a lot for your work,

Authors

Reviewer 2 Report

In this manuscript, the authors applied ultra-high-performance liquid chromatography (UPLC) and capillary gel electrophoresis (CGE) to investigate plasma and IgG N-glycome, respectively from obstructive sleep apnea (OSA) patients. 39 and 27 glycan peaks were observed in plasma and IgG sample, respectively. They found that three plasma and one IgG peaks were significantly elevated and decreased in patient samples. They also found that patients with severe OSA exhibited accelerated biological aging, with a median of 6.9 years more than their chronological age. Finally, the authors concluded that plasma N-glycome can be used as a biomarker for severe OSA.

Although the rationales and results seem straight forward, some major comments should be addressed.

Major comment:

1.      Because the mean apnea-hypopnea index (AHI), mean ages, and the mean BMI for patients were significantly higher than control, the relationships between AHI, ages, and BMI with altered glycans (as well as Glycan age) are not clarify in the current studies (even the authors used linear regression model with age, sex, and BMI to support their findings).

2.      Immunoglobulins accounts for 85-90% of plasma glycoproteins, and IgG accounts for 75-80% of total immunoglobulins. Thus, the patterns of N-glycan in plasma should be similar with IgG whether the patterns were analyzed by UPLC or CGE.

3.      It is suggested to apply comparable control cohort with case group.

4.      Chronic inflammation is one of the main pathogenic mechanisms of OSA. Does any chronic inflammation showed the same N-glycan patterns with OSA?

5.      Could the authors add glycan structure lists as the supplementary files?

Good.

Author Response

(The authors gave the same response as above.)

Round 2

Reviewer 1 Report

The authors have considered all my objections and answered or corrected the manuscript appropriately. The manuscript is now ready for publication.